# Planning Spatial Layout of a Typical Salt Tolerant Forage of Sweet Sorghum in the Yellow River Delta via Considering Resource Constraints, Nitrogen Use Efficiency, and Economic Benefits

**DOI:** 10.3390/plants12132483

**Published:** 2023-06-29

**Authors:** Yinan Gao, Changxiu Shao, Zhen Liu, Zhigang Sun, Buju Long, Puyu Feng

**Affiliations:** 1Institute of Geographic Sciences and Natural Resources Research, Chinese Academy of Sciences, Beijing 100101, China; sy20203030519@cau.edu.cn (Y.G.);; 2College of Resources and Environmental Sciences, China Agricultural University, Beijing 100193, China; 3Beijing Presky Technology Co., Ltd., Beijing 100195, China; 4College of Resource and Environment, University of Chinese Academy of Sciences, Beijing 100049, China; 5Zhongke Shandong Dongying Institute of Geography, Dongying 257000, China; 6College of Land Science and Technology, China Agricultural University, Beijing 100193, China; fengpuyu@cau.edu.cn

**Keywords:** saline-alkali land, medium and low yield fields, forage, nitrogen use efficiency

## Abstract

In order to effectively address the issue of severe soil salinization in the coastal area of the Yellow River Delta, which has led to a significant number of medium and low-yield fields in this region, and to satisfy the rising demand for feed grain in China in recent years, a highly effective solution is to replace conventional crops by cultivating a specialized type of forage grass that can withstand high salinity levels and is well adapted to the local climate. This study proposed a spatial layout scheme for planting salt-tolerant forages, with the aim of providing a foundation for enhancing saline-alkali land and increasing resource utilization efficiency. The results showed that the climate conditions in the Yellow River Delta were suitable for planting sweet sorghum. With respect to soil salt content, the suitable planting regions for sweet sorghum can be classified into four categories: Suitable, moderately suitable, less suitable, and unsuitable, with soil salt concentrations of 2.62–5.25‰, 5.25–7.88‰, respectively. Concerning economic benefits, sweet sorghum’s input-output ratio (74.4%) surpasses that of cotton in high saline-alkali zones, providing a significant advantage in comparison with traditional crops. In non-saline-alkali and light saline-alkali areas, the traditional winter wheat-summer maize planting system offers higher economic benefits and nitrogen use efficiency, so it is recommended to maintain this system as the dominant agricultural model. In moderately and severe saline-alkali zones, although one-season maize exhibits greater nitrogen efficiency, its economic benefits are lower than those of sweet sorghum. Hence, it is advisable to promote one-season maize in suitable regions and introduce salt-tolerant forage, such as sweet sorghum in other areas. This approach offers novel ideas and methods for crop spatial layout planning and addresses potential feed grain shortages in the region.

## 1. Introduction

The Yellow River Delta is among the newest deltas and emerging lands globally. However, like other coastal impact zones, the area has experienced severe soil salinization due to prolonged exposure to seawater, deep-seated groundwater, and climate changes [1,2,3]. Research has highlighted significant temporal and spatial variations in soil salinization in the Yellow River Delta [4]. In the past two decades, there has been a rapid and extensive increase in soil salinization from coastal to inland regions [5,6,7]. With a development history of just over 100 years, the Yellow River Delta primarily focuses on cultivating grain crops. However, soil salinization has led to the degradation of local farmland and ecosystems and has severely restricted the agricultural and economic development of the region [8]. Over the past few decades, various methods have been attempted to improve saline-alkali land, with plant restoration emerging as a reliable and low-cost technology. This approach involves planting salt-alkali-resistant plants to enhance soil quality and production potential [9,10]. The phenological period of non-salt-tolerant crops in the Yellow River Delta exhibits a delayed pattern from southwest to northeast, with more low-yield fields in the coastal areas and the grain yields per mu mostly below 100 kg [6]. Research indicates that selecting plants with different salt-alkali tolerance ranges to adapt to various saline-alkali soils is an effective method for improving and developing saline-alkali land [11]. Given the changing dietary habits of residents and their increasing nutritional and health needs, the current planting system cannot meet the rising demand for feed grain in rapidly growing livestock production [11,12,13,14]. Consequently, China’s food security has shifted toward feed grain security. Due to limited agricultural resources in China, exploring alternative farming methods to address the ever-grossing issue of forage scarcity is essential. A potential solution involves utilizing marginal lands, such as saline-alkali or low-quality soils, to avoid competition with food crops for land use [15]. Using food as feed equates to wasting other above-ground biomass of crops used as feed grains. Cultivating salt-alkali-tolerant forage can help to utilize the entire plant biomass and effectively improve the efficiency of resources [16]. As a result, scholars are currently focusing on developing a plan that matches the resource elements of the main suitable grasses for the Yellow River Delta’s coastal saline-alkali land in accordance with the screening requirements. This plan aims to adjust the agricultural planting structure, increase the area of artificially planted grasses to supplement the shortage of feed and establish an efficient agricultural system scheme that optimizes the utilization of climate resources.

Due to their strong ecological adaptability, high yield, and nutritional value, forage grasses from the *Poaceae* family—such as tall fescue, sweet sorghum, and ensilage maize—are the favored choices for ameliorating saline-alkali soils [17]. Among these, sweet sorghum is better suited for cultivation in the relatively humid climate conditions of eastern China compared to maize. Its nutritional value is higher, and it can be harvested multiple times, rendering it an attractive alternative to ensilage feed [18,19,20,21]. Both tall fescue and sweet sorghum have remarkable salt tolerance and exhibit significant soil improvement effects on coastal saline-alkali soils. However, planting sweet sorghum results in a greater enhancement in overall soil quality than planting tall fescue [22].

Sweet sorghum, belonging to the family *Poaceae* and genus *Sorghum*, is an annual herbaceous plant characterized by renewable, drought-resistant, waterlogging-resistant, salt-alkali-resistant, and high nitrogen fertilizer utilization [23,24,25,26]. Previous research has highlighted sweet sorghum’s exceptional adaptability to marginal soil conditions as well as its substantial biomass productivity and grass quality [27]. As a highly efficient C4 plant under short-day conditions, sweet sorghum offers high biomass and sugar content as forage, delivering an elevated level of total digestible nutrients [28]. Concerning its suitable distribution region, some scholars have conducted relevant research. Zhang et al. determined that sweet sorghum could be cultivated in most regions of China based on factors such as slope, average daily temperature, precipitation, and soil nutrients. The available land area compatible with sweet sorghum cultivation is as large as 5919.2 × 10^4^ hm^2^ [29]. The minimum natural environmental requirements for planting sweet sorghum include the accumulated temperature above 10 °C exceeding 2500 °C·d, soil pH between 5.0 and 8.5, slope limit of planting area not exceeding 25, and soil sand content below 85% [30]. However, most studies on the production potential of sweet sorghum are confined to statistical data and qualitative analysis, with a scarcity of quantitative research based on spatial data concerning the suitable distribution of sweet sorghum for planting in coastal saline-alkali regions. Moreover, the investigation of the economic significance of cultivating feed crops in developing countries reveals that the social benefits of planting forage surpass those of traditional crops [31]. Simultaneously, sweet sorghum and other crops have low nitrogen fertilizer requirements, which facilitate limiting fertilization rates, reducing environmental emissions, and providing favorable ecological benefits [32]. Consequently, sweet sorghum is an ideal grass for analysis in saline-alkali land in the Yellow River Delta.

Considering the natural conditions, production costs, and comprehensive output factors, is it more reasonable to cultivate sweet sorghum in the coastal saline-alkali land of the Yellow River Delta than traditional crops? There is currently a dearth of research on this subject. To address this gap, the present study aimed to analyze the primary crops grown in the Yellow River Delta region and examine the economic gains of sweet sorghum. Moreover, the study explored fertilizer utilization efficiency and assessed the feasibility of transitioning from food to feed crops in the region. Ultimately, the study will offer well-founded recommendations to the government concerning guiding structural adjustments in planting.

## 2. Results

### 2.1. Temporal and Spatial Distribution of Climate Suitability for Sweet Sorghum

Figure 1 illustrates the spatial distribution of climate suitability for sweet sorghum cultivation in the Yellow River Delta across three periods: 1980–1989, 1990–1999, and 2000–2014. Between 1980 and 2014, the temperature suitability for this crop in the region consistently exceeded 0.9, with the central area demonstrating marginally greater temperature compared to the southwestern and northeastern regions. The average temperature suitability for the periods 1980–1989 and 2000–2014 was 0.92 and 0.9, respectively, both higher than the 0.91 average observed during 1990–1999. Sweet sorghum precipitation suitability in the Yellow River Delta varied from 0.55 and 0.75 over the analysis period, with an overall regional average of 0.7 between 2000 and 2014. The central region typically displayed higher precipitation suitability than the southeastern and northwestern regions. The average precipitation suitability for the periods 1980–1989 and 2000–2014 were 0.69 and 0.70, respectively, both slightly greater than the 0.61 average recorded during 1990–1999. Regarding sunlight, the northern and eastern regions of the Yellow River Delta demonstrated marginally higher suitability than the southern and western regions. The average sunlight suitability for the periods 1980–1989 and 2000–2014 were 0.76 and 0.72, respectively, both above the 0.65 average noted during 1990–1999. Furthermore, sweet sorghum’s comprehensive suitability in the Yellow River Delta between 1980 and 2014 ranged from 0.70 to 0.81, indicating that the region can generally fulfill the climatic requirements for this crop’s cultivation. The northeastern region frequently exhibited higher comprehensive suitability compared to the southern and western regions. The average comprehensive suitability for the periods 1980–1989 and 2000–2014 were 0.78 and 0.75, respectively, both slightly greater than the 0.74 average observed during 1990–1999. Based on the climate suitability analysis, it can be concluded that a majority of areas in the Yellow River Delta meet the necessary water and heat resource requirements for successful growth and development of sweet sorghum.

### 2.2. Spatial Distribution of Soil Suitability for Sweet Sorghum

Figure 2 illustrates the distribution of saline-alkali land across the 19 counties of the Yellow River Delta, determined through a soil sampling survey conducted during spring 2020 (May–June). The results revealed a patchy spatial distribution of soil salinity within the Yellow River Delta, with considerable heterogeneity. Soil salinity values displayed a gradual decrease in a belt-shaped pattern from the river mouth to the south and west. Specifically, the central and northern areas of Dongying City and the coastal regions in the northern part of Binzhou City showed soil salinity values exceeding 2‰. Additionally, the soil salinity near the Yellow River’s mouth revealed values greater than 10‰.

The relationship between sweet sorghum yield and soil salinity was determined by fitting equation using field trial data obtained from the Kenli District of Dongying City. Based on the results, the distribution of soil suitability for sweet sorghum was acquired. Table 1 display the linear and polynomial fitting of the relationship between soil salinity and sweet sorghum fresh weight. The experimental outcomes revealed a clear negative correlation between sweet sorghum yield and soil salinity, irrespective of whether it was a linear or quadratic polynomial fitting (*p* < 0.05). The fitting outcomes allowed the experimental points to be categorized into four areas based on their yields, accounting for 75%, 50%, and 25% of the highest yield, from high to low: Suitable area, moderately suitable area, moderately unsuitable area, and unsuitable area (Table 2).

Interpolation of data obtained from two soil sampling surveys conducted in spring and autumn of 2020 produced a soil salinity distribution map of the Yellow River Delta (Figure 2). Further, random soil sampling and crop yield measurement were carried out to establish the relationship between soil salinity and sweet sorghum yield in field experiments. Based on the forage yield data obtained, the Yellow River Delta was divided into four regions according to their yields, making up 75%, 50%, and 25% of the highest yield. In the case of sweet sorghum cultivation, areas with soil salinity levels below 2.62‰ were classified as suitable, those with levels between 2.62‰ and 5.25‰ were moderately suitable, whereas regions with levels between 5.25‰ and 7.88‰ were deemed as moderately unsuitable, with a yield-to-highest-yield ratio ranging from 25% to 50%. Finally, zones with soil salinity levels above 7.88‰ were unsuitable for sweet sorghum cultivation (Figure 3).

### 2.3. Economic Benefit Analysis of Different Crops in the Yellow River Delta

A questionnaire survey in September 2021 to collect data on the production costs, economic outputs, and input-output ratios of the five major crops (wheat, maize, rice, soybean, and cotton) grown in the Yellow River Delta’s typical agricultural regions in Dongying, Shandong Province. The objective of the study was to suggest reasonable crop planting plans by comparing the economic benefits of various crops (Table 3).

The results indicated that the wheat-millet double-season planting system had the highest input-output ratio in non-saline-alkali and mildly saline-alkali areas, which was far higher than that of other crops. Soybean had the lowest input-output ratio and economic output among all the crops in both non-saline-alkali and mildly saline-alkali areas. Maize had an input-output ratio of 178.06% and 161.65% in non-saline-alkali and mildly saline-alkali areas, respectively, while wheat had ratios of 161.52% and 136.03% in these two regions, respectively.

Based on experimental plot data, sweet sorghum in non-saline-alkali regions had an economic output of 17,712.15 CNY/ha and an input-output ratio of 215.8%. In areas with low soil salinity, sweet sorghum was shown to have higher economic benefits compared to traditional crops, making it an attractive choice for farmers. In moderately and heavily saline-alkali areas, rice was the main crop, with an economic output of 17,598 CNY/ha and 16,500 CNY/ha, and input-output ratios of 137.46% and 128.87%, respectively. Cotton has also gained popularity among farmers in high soil salinity areas, but high labor costs have resulted in input-output ratios below 1, even with high yields (an economic output of 14,850 CNY/ha and an input-output ratio of 43.11% in heavily saline-alkali areas).

Overall, the study recommended that farmers choose crops based on salinity levels and the economic benefits in particular regions. For areas with low soil salinity, sweet sorghum was considered to have great potential for high economic returns, whereas rice remained the main crop in heavily saline-alkali areas.

### 2.4. Fertilizer Utilization Rate in a Typical Agricultural Area of the Yellow River Delta

The study categorized cropping systems into five categories: Double-season wheat, double-season maize, one-season cotton, one-season rice, and one-season maize. Effective questionnaires were obtained from each category, with 38 responses for double-season wheat and double-season maize, 23 responses each for one-season cotton and one-season maize, and 10 responses for one-season rice. To determine optimal fertilization rates for different crops, quadratic regression was performed on fertilization rates per unit area and crop yields per unit area, which provided non-fertilization yields, maximum yields, and the fertilization rate at maximum yield. The non-fertilization yields (control yields) for double-season wheat, double-season maize, one-season cotton, one-season rice, and one-season maize were 4.294, 5.800, 2.106, 6.440, and 6.366 Mg/ha, respectively. Based on the survey results, I organized and calculated the fertilizer data used by farmers, including the application rates of nitrogen (N), phosphate (P_2_O_5_), potassium (K_2_O), and the total amount of fertilizer applied by aggregating these three nutrients. Among the different crops, double-cropping wheat required the highest amount of fertilizer to reach maximum yields, at 0.995 Mg/ha, consisting of 0.444 Mg/ha of nitrogen, 0.515 Mg/ha of phosphate, and 0.036 Mg/ha of potassium. The maximum yield of double-cropping maize was obtained with 0.447 Mg/ha of fertilizer, including 0.220 Mg/ha of nitrogen, 0.208 Mg/ha of phosphate, and 0.019 Mg/ha of potassium. The lowest total amount of fertilizer required to reach the maximum yield of single-cropping cotton was 0.342 Mg/ha, comprising 0.139 Mg/ha of nitrogen, 0.153 Mg/ha of phosphate, and 0.049 Mg/ha of potassium. For single-cropping rice, the maximum yield required 0.540 Mg/ha of fertilizer, containing 0.279 Mg/ha of nitrogen, 0.208 Mg/ha of phosphate, and 0.019 Mg/ha of potassium, while single-cropping maize reached the maximum yield with 0.829 Mg/ha of fertilizer, including 0.383 Mg/ha of nitrogen, 0.337 Mg/ha of phosphate, and 0.0109 Mg/ha of potassium.

In the southern part of Dongying City, encompassing non-saline-alkali areas, mildly, and moderately saline-alkali areas, the primary cropping system is the winter wheat-summer maize rotation. Conversely, the northern part of Dongying City, characterized predominantly by moderately and heavily saline-alkali zones, employs one-season maize, one-season rice, and one-season cotton cropping system as the principal agricultural practice. The fertilizer utilization efficiency of winter wheat and summer maize within the winter wheat-summer maize rotation system declines as soil salinity increases. The mean fertilizer utilization rates of winter wheat in non-saline-alkali, mildly saline-alkali, and moderately saline-alkali double cropping areas were 24.9%, 27.2%, and 18.9%, respectively, while the corresponding nitrogen utilization rates were 33.0%, 27.1%, and 11.8%. Similarly, the average fertilizer utilization rates of summer maize in non-saline-alkali, mildly saline-alkali, and moderately saline-alkali double cropping areas were 38.2%, 34.4%, and 17.8%, respectively, with the respective nitrogen utilization rates being 38.2%, 34.4%, and 17.8% (Figure 4).

## 3. Discussion

### 3.1. Planting Suitability of Sweet Sorghum

Sweet sorghum has flexible climate requirements for growth, demanding cumulative temperatures of over 2500 °C·d and a daily average temperature above 10 °C with a precipitation range of 400–1000 mm [30]. These climate conditions are conducive to the growth of sweet sorghum in the Yellow River Delta. By employing crop temperature suitability, precipitation suitability, and light suitability calculations, the climate suitability of sweet sorghum in the Yellow River Delta was assessed. Between the years 1990 and 1999, there was a significant decrease in precipitation suitability due to reduced rainfall in the area. However, precipitation levels gradually recovered after 2000 [33]. Despite this, sweet sorghum exhibits high drought tolerance [34,35] and requires lower water during the growth and development stages [36]. Moreover, sweet sorghum has higher water-use efficiency than other summer crops when water is scarce [37]. Although there has been a slight reduction in photoperiod suitability since 2000, sweet sorghum possesses natural mechanisms to protect its own photosynthetic system, promote sucrose biosynthesis, and inhibit the degradation of chlorophyll [26]. Therefore, climatic factors in the Yellow River Delta are not the primary limiting factors affecting the growth of sweet sorghum.

A significant negative correlation was observed between sweet sorghum yield and soil salinity, irrespective of the fitting method (Table 1). Based on the threshold values for dividing the suitability of planting based on soil salinity, the distribution of suitability for sweet sorghum planting in the Yellow River Delta was subdivided, as shown in Figure 3. Consistent with Vasilakoglou’s findings, sweet sorghum can grow normally in soil salinity of 3.2 dSm^−1^. Growth is adversely affected when soil salinity exceeds this value [24]. This is primarily because, under salt-stress conditions, sorghum can regulate permeability by transporting ions to vacuoles or accumulating soluble substances via the barrier function of its root apoplast [38]. Comprising Casparian bands and lignin layers, the apoplast can block the extracellular transpiration bypass of Na^+^ [39,40], while salt excretion can accumulate Na^+^ in the roots and restrict transport to the stem [41]. In addition, the dynamic balance of reactive oxygen species (ROS) content within the plant is critical for growth. When ROS accumulation surpasses cellular tolerance, it can damage the plant’s cell structures or facilitate cell apoptosis [42]. Sweet sorghum can adapt to the secondary oxidative stress caused by salt stress by clearing ROS via enzymatic and non-enzymatic antioxidant systems [43]. Moreover, sweet sorghum can grow in moderately to slightly saline-alkali soil. Ultimately, it proves that the distribution of sweet sorghum planting in the Yellow River Delta is primarily constrained by soil salinity, showing a distinct belt-like distribution of planting suitability. Unsuitable areas for growing sweet sorghum mainly correspond to high-salinity coastal zones in the northeast of the Yellow River Delta.

### 3.2. Comparison of Economic Benefits between Main Crops and Sweet Sorghum in the Yellow River Delta

In non-saline-alkali areas, the total output of maize and wheat reached 18,859.5 CNY/ha and 19,047 CNY/ha, respectively, and their input-output ratios were the highest among the five economic crops currently grown, at 178.06% and 161.52%, respectively. Although cotton yielded high economic returns in both non-saline-alkali areas and mildly saline-alkali areas, its input-output ratio was the lowest of all crops due to labor input costs. Sweet sorghum had an output of 17,712.15 CNY/ha in non-saline-alkali areas, and its labor input costs were much lower than those of traditional crops. Liu et al. Found that sweet sorghum has a lower input cost and a higher input-output ratio than cotton [44].

Hariprasanna and Rakshit argued that sorghum cultivation requires minimal external inputs, such as fertilizers and plant protection measures [45]. As a result, sweet sorghum has a higher input-output ratio than maize and wheat. However, its economic returns are still noticeably lower than traditional crops, which aligns with the research findings of Chapke and Tonipi [46]. In mildly saline-alkali areas, the economic output of sweet sorghum is significantly affected by soil salinity, and its input-output ratio is lower than that of maize and wheat. Consequently, it is advised to maintain the existing system in areas where winter wheat and summer maize can be grown without salt-alkali issues.

In moderately and heavily salt-alkali areas, the economic output of wheat decreased significantly, to only 11,223 CNY/ha and 9960 CNY/ha, respectively, with maize’s economic output also decreasing. Consequently, the winter wheat-summer maize rotation system is significantly affected by soil salinity, rendering it unsuitable for large-scale planting. Cotton and rice are more salt-alkali tolerant crops: The economic output and input-output ratio of rice in moderately and heavily salt-alkali areas both exceeded those of sweet sorghum. However, large-scale rice planting necessitates a substantial volume of irrigation water, which is unfeasible under limited irrigation conditions. In recent years, human activities and climate change have led to a significant reduction in water flow in the lower reaches of the Yellow River [47], leaving insufficient water resources to support the ecological system of the Yellow River Delta [48]. Cotton planting warrants high labor input costs [49] and substantive amounts of fertilizers, which exacerbate fertilizer waste and non-point source pollution in the Yellow River Delta, counteracting soil and environmental improvement measures [50]. Analyzing the suitability of sweet sorghum planting demonstrates that sweet sorghum yield markedly decreased in moderately and heavily salt-alkali soils, with yield reduction exceeding 50% in areas where the soil salinity is between 4.45‰ and 6.67‰. Nevertheless, the input-output ratio remained high. Thus, it is recommended to expand the planting of sweet sorghum in this region. The integration of mechanization, optimized variety selection, and production techniques can reduce labor and fertilizer costs while increasing gross profits from sweet sorghum cultivation. This approach offers significant economic potential for the development of the sweet sorghum industry [44]. Especially in light of global climate change, sorghum has the opportunity to become an important food and cash crop [51]. Additionally, due to its fast growth, high yield, high dry matter content, leafiness, wide adaptability, and drought tolerance, sorghum is considered an essential feed [45].

### 3.3. Comparative Analysis of Nitrogen Efficiency between Sweet Sorghum and Traditional Cropping Systems

A comparative analysis of the fertilizer utilization rates of sweet sorghum and other major crops in different saline-alkali areas, based on field experiments and farmer surveys, shows:

The fertilizer utilization efficiency of crops in the Yellow River Delta region is generally low, attributable to the detrimental effects of soil salinity on rhizobia [52], host plants [53], and their symbiotic relationship [54,55]. Excessive soil salt levels can reduce nitrogen-fixing activities of plants, leading to reduced fertilizer utilization efficiency in crops [56].

In non-saline-alkali areas, the nitrogen fertilizer utilization rates of winter wheat and summer maize are as high as 33.0% and 31.0%, respectively (Figure 5), slightly lower than the nitrogen fertilizer utilization efficiency of sweet sorghum. Sweet sorghum effectively absorbs nitrogen from the soil due to its root morphology [25]. Sorghum roots exhibit mycorrhizal association, permitting direct absorption of organic nitrogen from the soil as compared to maize [57,58]. Additionally, some scholars have found that sorghum roots can exude secondary metabolites that function as nitrification inhibitors in the soil, retaining fertilizers in the form of NH_4_-N for more extended periods. This increases crop nitrogen uptake and utilization efficiency while reducing N losses caused by leaching and denitrification [59,60]. However, the annual yield of winter wheat and summer maize, producing two crops per year, surpasses that of other cropping systems. Therefore, in non-saline-alkali areas, it is advisable to maintain the winter wheat-summer maize planting system as the dominant agricultural model.

In slightly saline-alkali areas, the nitrogen fertilizer utilization rates of winter wheat and summer maize decreased, similar to the nitrogen fertilizer utilization efficiency of cotton. Although cotton has a higher nitrogen fertilizer utilization rate in lightly saline-alkali areas, its long growing period (from April to September) necessitates multiple pesticide applications and other management measures. As labor input cost increases year by year, cotton planting has gradually been abandoned by farmers. The nitrogen fertilizer utilization rate of one-season maize in the study area is 35.4%. Its growing period (from May to August) coincides with when rainfall is concentrated, causing maize plants to be severely weakened by soil salinity stress due to rainwater leaching. Hence, it is recommended to maintain the dominant winter wheat-summer maize planting system and encourage one-season maize planting in slightly saline-alkali areas.

In moderately to heavily saline-alkali areas, the winter wheat-summer maize planting system is most significantly affected by soil salinity and is not viable in some areas. Although cotton has a nitrogen fertilizer utilization efficiency similar to sweet sorghum, it has been gradually replaced by other crops in recent years due to management measures and agricultural input factors. The nitrogen fertilizer utilization rate of one-season maize is also significantly lower, likely due to the heavy soil salinity in the region, which does not lower the soil salinity to a level required by maize even during the rainy season, resulting in considerable differences in maize yields. The growth of rice is constrained by irrigation conditions, and while the planting area has expanded in recent years, the overall cultivated area remains relatively small. Sweet sorghum boasts a significant advantage in nitrogen fertilizer utilization rate in moderately to heavily saline-alkali areas, with rates of 20.9% and 19.6%, respectively, which are stable and higher than those of other cropping systems. Moreover, sweet sorghum is highly efficient at retrieving fertilizer N [19]. Therefore, it is recommended that the cultivation of saline-tolerant forage crops, such as sweet sorghum, be promoted in moderately to heavily saline-alkali areas.

## 4. Materials and Methods

### 4.1. Study Area

The Yellow River Delta High-Efficiency Ecological Economic Zone (hereafter referred to as the Yellow River Delta) encompasses all counties, cities, and districts within Dongying and Binzhou cities in Shandong Province, as well as select districts and counties in the neighboring Weifang, Dezhou, Zibo, and Yantai cities, totaling 19 counties and districts (Figure 6). The entire research area covers 2.65 × 10^4^ hm^2^. Characterized by a warm temperate semi-humid continental monsoon climate, the Yellow River Delta experiences an average annual temperature of 12.4 °C, annual precipitation of 551.6 mm, and average annual evaporation of 1928.2 mm [33]. With an annual accumulated temperature of ≥0 °C equal to 4713.5 °C·d, and ≥10 °C equal to 4245 °C·d, coupled with an average annual sunshine duration of 2629 h, the region boasts favorable light and heat conditions, rendering it suitable for agricultural development. Land use data for the Yellow River Delta were obtained from GlobeLand30 data, a crucial milestone of the global land cover remote sensing mapping and key technology research project under the Chinese national high-tech research and development program (863 program).

### 4.2. Data Source

Meteorological data, which include daily temperature and precipitation data from 1980–2014, were sourced from the China Meteorological Science Data Sharing Service Network (http://data.cma.cn/, accessed on 10 September 2020) and cover 19 meteorological stations within and surrounding the Yellow River Delta. Daily average temperature and precipitation were calculated based on crop growth periods to determine temperature and precipitation suitability, followed by spatial interpolation using the Inverse Distance Weighted method (IDW).

Soil data were derived from a soil sampling survey conducted from May to June 2020 across 19 districts and counties within the Yellow River Delta. Initial points were established using a 10 km grid, taking into account factors, such as water bodies, building distribution, surface complexity, and plot distribution. Leveraging the 2019 Sentinel-2 remote sensing image, 121 points were selected for sampling surveys. Soil samples were collected concurrently with crop samples at a depth of 0–10 cm, 10–20 cm, and 20–40 cm. Laboratory analyses were conducted for soil moisture content, pH, and electrical conductivity. Soil salinity was determined by measuring the electrical conductivity of saturated slurry and the relationship between conductivity and salt content. Spatial interpolation of soil salinity in the 0–20 cm layer was undertaken using inverse distance weighting to derive the spatial distribution map of soil salinity. Sweet sorghum yield data was acquired from the grassland trials at the Chinese Academy of Sciences Shandong Dongying Research Institute, where KeTian10 and KeTian2 varieties were planted according to the gradient of soil salinity. At the site, 58 random sampling points were selected for soil salinity and yield measurements. Sweet sorghum yield was ascertained by harvesting a 1 m^2^ plot of sweet sorghum at each observation point and measuring the fresh weight to estimate the yield per mu. Crop survey data were sourced from typical agricultural households in the core area of the Yellow River Delta, primarily for economic analysis of grasslands and traditional crops and fertilizer utilization efficiency. Production input costs include seed, fertilizer, pesticide, labor, and harvesting expenses, while the total output value is the product of crop yield and the local purchase price for the corresponding year.

### 4.3. Climate Suitability for Sweet Sorghum

The primary factors influencing sweet sorghum growth include temperature, sunshine duration, and precipitation. Therefore, this study assessed the suitability of temperature, sunshine, and water for sweet sorghum at various growth stages [61].

#### 4.3.1. Temperature Suitability Index

Temperature suitability represents a gauge of air temperature for crop growth and development during each growth stage, with the most suitable value being 1 and the least suitable value being 0. The calculation method for the beta function is as follows:(1)$$F\left(T\right)=\left\{\begin{array}{cc} \frac{\left(T-T_{l}\right){\left(T_{h}-T\right)}^{B}}{\left(T_{o}-T_{l}\right){\left(T_{h}-T_{o}\right)}^{B}} & T_{l}<T<T_{h}\\ 0 & T\geq T_{h}\;or\;T<T_{l} \end{array}\right.\;B=\frac{T_{h}-T_{o}}{T_{0}-T_{l}}$$
where, F(*T*)—temperature suitability when the actual temperature is *T*, *T_o_*—crop’s optimal growth temperature, *T_l_*—crop’s minimum growth temperature, below which the development rate is 0, *T_h_*—the crop’s maximum growth temperature, above which the development rate stops. *B*—constant calculated from the three base temperatures of each growing period.

#### 4.3.2. Water Suitability Index

Water suitability denotes a measure of precipitation appropriateness for crop growth and development during each growth stage, with values ranging from 0 to 1. Precipitation serves as the principal source of water for crops and soil, and it is intimately linked with crop growth, development, and yield formation. The research findings indicate that typical crop water requirements can be employed as a benchmark for suitable water for crop growth. Mild drought is considered when precipitation during the growth period is less than 30% of the water requirements, while mild waterlogging is considered when it exceeds 30% of the water requirements. Consequently, the precipitation coefficients for mild drought and mild waterlogging are set as 0.7 and 1.3, respectively.
(2)$$F\left(R\right)=\left\{\begin{array}{c} 1\;R_{b}\leq R\leq R_{m}\\ R/R_{b}\;R<R_{b}\\ R_{m}/R\;R>R_{m} \end{array}\right.\;R_{b}=0.7\times {ET}_{c}\;R_{m}=1.3\times {ET}_{c}$$
where F(*R*)—precipitation suitability of crops, *R*—actual cumulative precipitation, *R_b_*—precipitation coefficient for mild drought, *R_m_*—precipitation coefficient for mild waterlogging, *ET_c_*—crop’s water requirements.

Water requirements pertain to the specific crop field evapotranspiration under optimal soil moisture and fertilizer conditions, where the crop develops normally, remains disease-free, and achieves a high-yield level. The Penman–Monteith method recommended by the Food and Agriculture Organization (FAO) is used for calculation [62,63].
(3)$${ET}_{c}={ET}_{0}\times K_{c}\;{ET}_{0}=\frac{0.408∆\left(R_{n}-G\right)+\gamma \frac{900}{T+273}u_{2}\left(e_{s}-e_{a}\right)}{∆+\gamma \left(1+0.34u_{2}\right)}$$
where, *ET_C_*—crop water requirements (mm/d), *K_C_*—crop coefficient, with values of 0.45, 1.15, and 1.05 for the sweet sorghum planting-emergence stage [64], flowering-grain filling stage, and maturity stage, respectively, *ET*_0_—reference crop evapotranspiration (mm/d), ∆—slope of the relationship between saturated vapor pressure and temperature curve (kPa/K), *R_n_*—net surface radiation (MJ/(m^2^·d)), *G*—soil heat flux (MJ/(m^2^·d)), which is negligible in the calculation of evapotranspiration in the model, *γ*—psychometric constant (0.067 kPa/°C), *u*_2_—daily average wind speed at a height of 2 m (m/s), *T*—daily average temperature (°C), *e_s_*—saturated vapor pressure (kPa), *e_a_*—actual vapor pressure (kPa).

#### 4.3.3. Sunshine Suitability Index

Research has shown that the flag emergence-grain filling stage is the sole photoperiod-sensitive period in the growth process of sweet sorghum, with other growth stages being insensitive to day length [65]. The development rate is influenced exclusively by temperature.
(4)$$F\left(S\right)=\left\{\begin{array}{c} S/S_{o}\;S<S_{o}\\ 1\;S\geq S_{o} \end{array}\right.$$
where F(*S*)—sunshine suitability, *S*—actual sunshine duration, *S_o_*—0.7 astronomical day length (available sunshine hours).

#### 4.3.4. Comprehensive Suitability Index

Drawing upon existing research results, this article classifies sweet sorghum into three growth stages and determines the climate suitability index threshold values for each stage [66,67] (Table 4).
(5)$${F\left(T,R,S\right)}_{i}=\sqrt[{3}]{{F\left(T\right)}_{i}\cdot {F\left(R\right)}_{i}\cdot {F\left(S\right)}_{i}}$$
where *i* = 1, 2, 3, 4, 5—planting-emergence stage, emergence-heading stage, heading-flowering stage, flowering-maturity stage, and maturity-harvesting stage, respectively.

### 4.4. Calculation of Fertilizer Utilization

Apparent crop recovery efficiency of applied nutrient (RE) pertains to the increased nutrient uptake in the above-ground crop components per unit of nutrient applied. This metric is currently the most prevalent expression for nutrient utilization efficiency utilized by researchers. The calculation formula is detailed below [68]:(6)$$RE=\frac{U-U_{0}}{F}\times 100\%$$
where, *U*—total nutrient uptake in the above-ground biomass of crops after fertilization, *U*_0_—total nutrient uptake in the above-ground biomass of crops under no fertilizer conditions, *F*—amount of fertilizer input.

The specific calculation method [69,70] for fertilizer agronomic efficiency is as follows:

Generally, the relationship between fertilizer application rate and grain yield follows a diminishing rate of return, in which crop yield increases with the growth of nitrogen fertilizer utilization rate within a particular range. After surpassing the critical value, crop yield decreases as the nitrogen fertilizer utilization rate increases [70]. By utilizing the data on fertilizer application rate per unit area and crop yield per unit area from 1991 to 2020, a parabolic equation was fitted:(7)$$Y=b_{0}+b_{1}X+b_{2}X^{2}$$
where *Y*—grain yield per unit area (Mg/ha), *X*—fertilizer application rate per unit sowing area (Mg/ha). According to the fitted equation, calculate the corresponding control yield, the maximum yield (the highest yield that can be achieved by fertilization under current conditions), and the fertilizer application rate corresponding to the maximum yield.

The yield with fertilizer is calculated by averaging the current and previous year’s yield as the current yield, then subtracting the yield without fertilizer. Subsequently, the nutrient associated with the current fertilizer application rate is calculated based on the nitrogen, phosphorus, and potassium content in the grain and straw [71,72]. Fertilizer utilization efficiency equals the percentage of nutrient uptake corresponding to the yield with fertilizer in relation to fertilizer input.

## 5. Conclusions

This study examined the natural conditions necessary for the growth of sweet sorghum, focusing on two aspects: Climatic suitability and soil suitability. Drawing upon an input-output survey conducted in the core area of the Yellow River Delta and field experiment data, the economic benefits were compared. The Yellow River Delta exhibits high climatic suitability for sweet sorghum, making the region appropriate for its cultivation. An analysis of experimental data reveals that the areas unsuitable for sweet sorghum cultivation are primarily located in the northeast, where the soil salt content is elevated. In non-saline-alkali regions, a typical double-season planting system of wheat and maize has been established, which not only ensures the country’s adequate food supply but also yields higher economic returns. Consequently, it is advisable to maintain the current situation in non-saline-alkali zones. From moderately to heavily saline-alkali areas, promoting the planting and the cultivation of sweet sorghum in regions with increased soil salinity is recommended in order to enhance both economic and environmental benefits.

## Figures and Tables

**Figure 1 plants-12-02483-f001:**
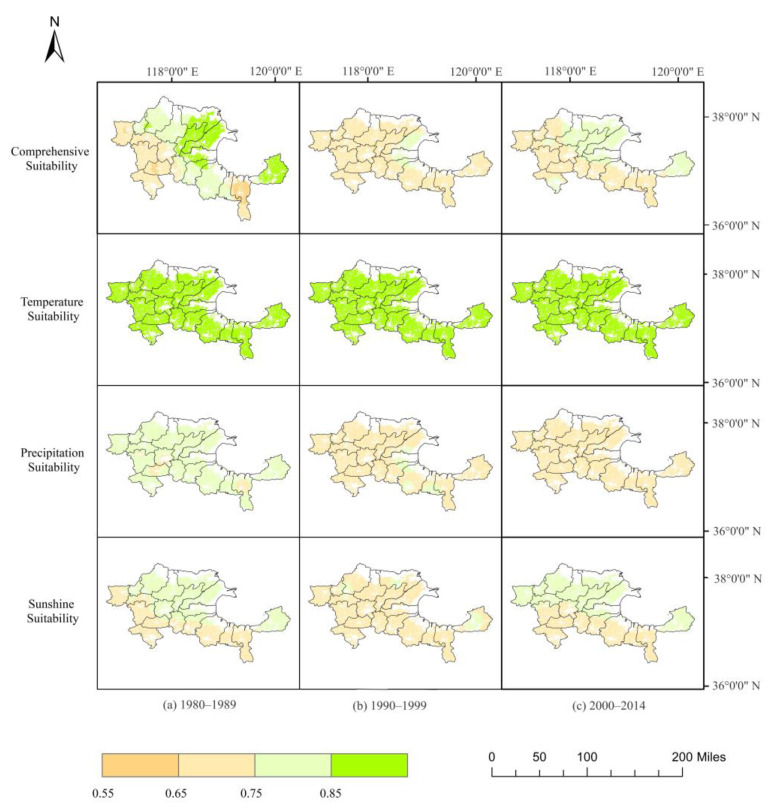
Spatial distribution of sunshine suitability, precipitation suitability, temperature suitability, and comprehensive suitability for sweet sorghum cultivation in cultivated land of the Yellow River Delta region at three temporal stages: (**a**) 1980–1989, (**b**) 1990–1999, and (**c**) 2000–2014.

**Figure 2 plants-12-02483-f002:**
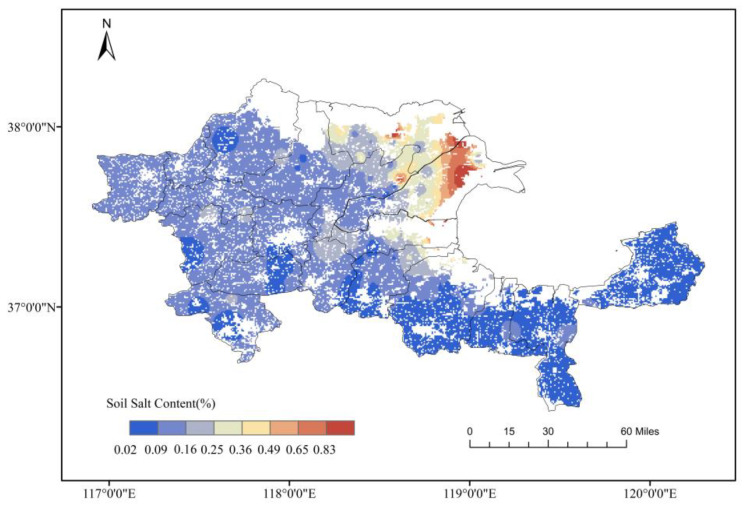
Spatial distribution of soil salinity in cultivated land of the Yellow River Delta region based on Inverse Distance Weighted (IDW) interpolation using soil sampling and measurement results.

**Figure 3 plants-12-02483-f003:**
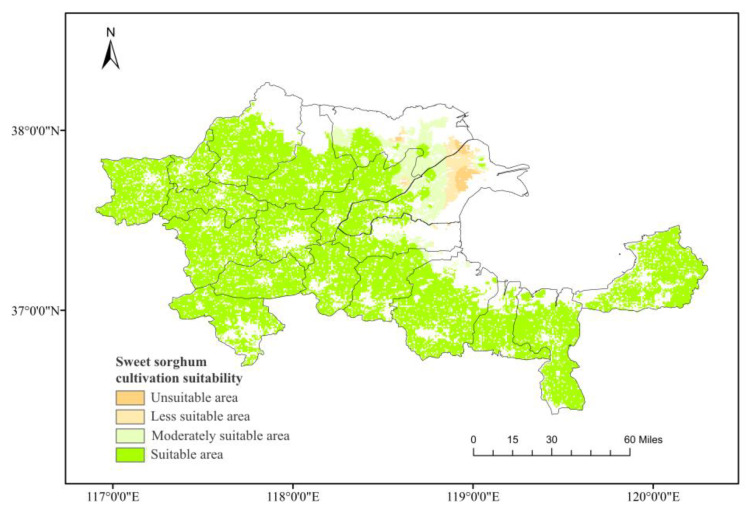
Classification of cultivated land in the Yellow River Delta region for sweet sorghum cultivation suitability into four categories based on the regression of soil salinity and corresponding sweet sorghum yield: Suitable region, moderately suitable region, less suitable region, and unsuitable region.

**Figure 4 plants-12-02483-f004:**
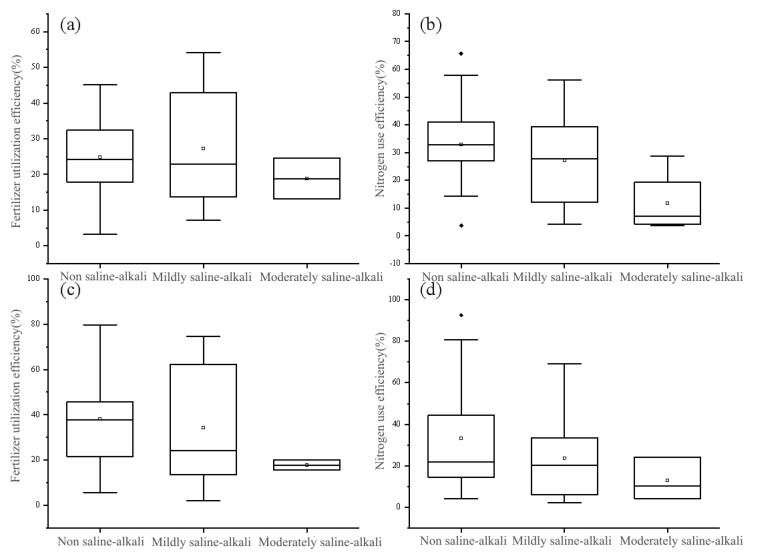
Fertilizer utilization efficiency (**a**,**c**) and nitrogen fertilizer utilization efficiency (**b**,**d**) of winter wheat and summer maize in double cropping area of Dongying City in 2020: (**a**,**b**) Winter wheat, (**c**,**d**) summer maize. Boxplots show the median and the 25th and 75th percentiles. Note: The little squares represent the mean of the data, while the black dots indicate outliers.

**Figure 5 plants-12-02483-f005:**
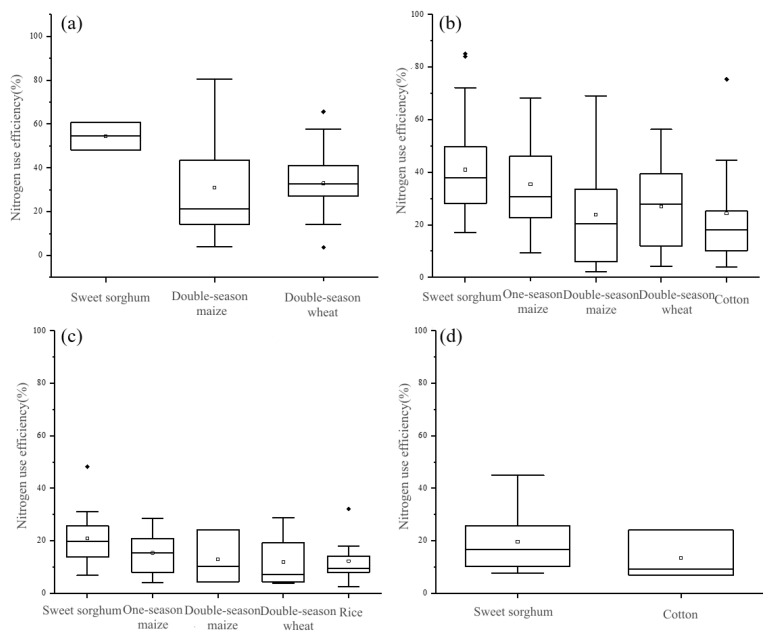
Comparison of nitrogen fertilizer efficiency of sweet sorghum and typical crops in different saline areas: (**a**) Non-saline, (**b**) mildly saline, (**c**) moderately saline, (**d**) heavily saline. Boxplots show the median, and the 25th and 75th percentiles. Note: The little squares represent the mean of the data, while the black dots indicate outliers.

**Figure 6 plants-12-02483-f006:**
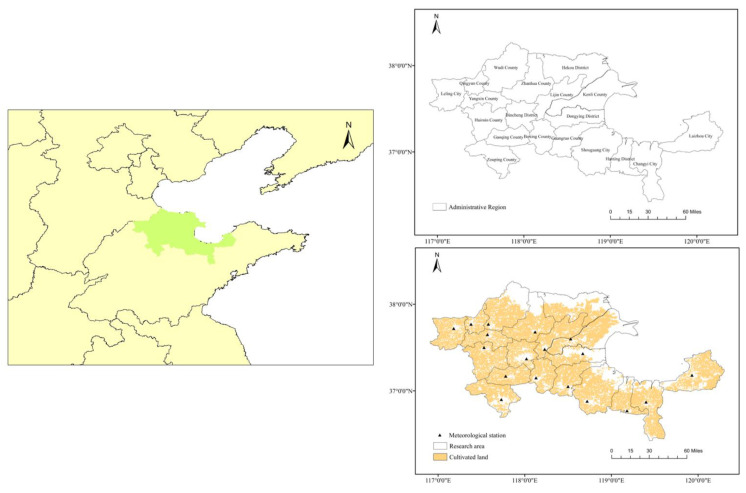
Plots depicting the location of the study site. Note: Map of the study area location (**left**), administrative divisions in the Yellow River Delta region (**upper right**), and meteorological stations and cultivated land distribution in the Yellow River Delta region (**lower right**).

**Table 1 plants-12-02483-t001:** Parameter values of linear and polynomial regression equations, coefficient of determination, and *p*-values for soil salinity and corresponding sweet sorghum yield.

Mode of Fitting	Equation	R^2^	*p*-Value
Linear fitting	y = −0.5077x + 5.3384	0.3763	*p* < 0.05
Polynomial fitting	y = 0.0697x^2^ − 1.2116x + 6.8317	0.4148	*p* < 0.05

**Table 2 plants-12-02483-t002:** Rates of yield to salinity-free stress condition sweet sorghum yield and corresponding soil salinity ranges for different categories of cultivated land in the Yellow River Delta region determined by regression of soil salinity and corresponding sweet sorghum yield.

Suitable Level	Suitable Area	Moderately Suitable Area	Less Suitable Area	Not Suitable Area
Ratio of yield to maximum yield (%)	≥75	50–75	25–50	≤25
Soil salt content (‰)	≤2.62	2.62–5.25	5.25–7.88	≥7.88

**Table 3 plants-12-02483-t003:** Comparison of input-output ratio (yield/input per hectare) and economic output per hectare of sweet sorghum and typical local crops in different areas with four levels of soil salinity in the Yellow River Delta region.

Crop Species	Degree of Soil Salinity							
	Non Saline-Alkali		Mildly Saline-Alkali		Moderately Saline-Alkali		Heavily Saline-Alkali	
	Input-Output Ratio (%)	Economic Output (CNY/ha)	Input-Output Ratio (%)	Economic Output (CNY/ha)	Input-Output Ratio (%)	Economic Output (CNY/ha)	Input-Output Ratio (%)	Economic Output (CNY/ha)
Soybean	82.78	8400	71.73	7278	/	/	/	/
Cotton	67.55	23,265	71.72	24,703.5	57.92	19,950	43.11	14,850
Rice	/	/	/	/	137.46	17,598	128.87	16,500
Wheat	161.52	19,407	136.03	16,041	95.17	11,223	/	/
Maize	178.06	18,859.5	161.65	17,121	158.59	16,797	/	/
Sweet sorghum	215.80	17,712	147.32	9723	90.41	5967	74.4	4911

Note: The symbol ‘/’ denotes un-planted or unrecorded data within the specified area.

**Table 4 plants-12-02483-t004:** Value of temperature, light hours, and crop coefficient indexes of sweet sorghum in three growth periods.

Developmental Stage	Emergence-Heading	Heading-Flowering	Flowering-Maturity
Date	From mid-May to mid-July	From mid-July to late August	From late August to early October
Upper temperature limit/°C	42	42	42
Lower temperature limit/°C	11	11	5.7
Optimal temperature/°C	30	30	22.5
B	0.63	0.63	1.16
The most suitable sunshine duration		11	
The upper limit of sunshine duration		13.6	
Kc	0.45	1.15	1.05

## Data Availability

The data presented in this study are available on request from the corresponding author.

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
