# Peer review of "Planning Spatial Layout of a Typical Salt Tolerant Forage of Sweet Sorghum in the Yellow River Delta via Considering Resource Constraints, Nitrogen Use Efficiency, and Economic Benefits"

_plants, 2023, doi:10.3390/plants12132483_

Round 1
Reviewer 1 Report
Plants 2390395
Planning spatial layout of a typical salt tolerant forage of sweet sorghum in the Yellow River Delta via considering resource constraints, nitrogen use efficiency, and economic benefits
Yinan Gao, Changxiu Shao, Zhen Liu, Zhigang Sun, Buju Long, Puyu Feng
The manuscript is interesting and reports novel date for the area of study. However, the following comments and concerns must be addressed before acceptance:
1. There are some typos found throughout the manuscript. Some have been indicated in the PDF file. Authors are urged to review the whole document and fix such issues.
2. Some other technical issues were detected in the PDF file; please revise and correct accordingly.
3. Though the results are conclusive, authors are asked to explain their findings in a deeper way. Why was sweet sorghum suitable? Which biochemical, physiological and molecular mechanisms may be used by the species to display such behavior? Please deeply explain your results and give come evidences as compared to other grasses.
4. The Reference list must be revised and corrected according to the editorial policies of the journal.
Other comments and concerns are indicated in the PDF file attached to this evaluation.

There are some typos found throughout the manuscript. Some have been indicated in the PDF file. Authors are urged to review the whole document and fix such issues.
Reviewer 2 Report
Comments and suggestions for Authors
Title: Planning spatial layout of a typical salt tolerant forage of sweet
sorghum in the Yellow River Delta via considering resource constraints, nitrogen use efficiency, and economic benefits
The subject is very interesting and correspond to the journal’s profile. The presented manuscript deals with the current local problem.
General value of the paper is very good. Article contains new information.The experimental dataset undoubtedly are useful and constitutes scientific values.
The aims of the study were to analyze the primary crops grown in the Yellow River Delta region and examine the economic gains of sweet sorghum, and fertilizer utilization efficiency, and investigates the feasibility of shifting from food to feed crops in the region.
The Results and Discussion section is well written. The research results are clearly presented and well described. References are well matched. The conclusions are correct.
General remarks
In order to increase the usefulness of the article, Authors must refer to the following points. Additions should be made to increase the scientific value of the manuscript.
1. Why is soil salinity expressed in ‰ throughout the manuscript? This unit refers to the salinity of water bodies. Please verify this unit.
2. Lines 213-214 Yields of test plants should be given in t ha-1 or Mg ha-1.
3. Lines 214-215 Specify the fertilizers used and the doses of individual fertilizer ingredients.
4. The low values of the utilization factors of nutrients from fertilizers should be explained.
5. The description of Figures 1-6 should be made more readable.
Specific comments
Line 142 – should be: Table 1 and 2…..
Line 435 – should be: ..[41,42].
Line 454 – should be: ..[45,46]…
References should be corrected according to publishing requirements.
Best regards
Minor editing of English language required.
Round 2
Reviewer 1 Report
The manuscript is in general well written, though moderate editing of English language is required. For instance, words like “area”, “popularize”, improving”, etc., are either misused or overused, leading to cacophony.
Some grammatical tenses are misused; for instance “are popularize” …
Figures are not clear and neat enough and must be re-edited and improved.
Titles in Tables and Legends in figures must be substantially improved. In general titles and legends must contain all information needed to be explained by themselves.
The manuscript is in general well written, though moderate editing of English language is required. For instance, words like “area”, “popularize”, improving”, etc., are either misused or overused, leading to cacophony.
Some grammatical tenses are misused; for instance “are popularize” …
Figures are not clear and neat enough and must be re-edited and improved.
Titles in Tables and Legends in figures must be substantially improved. In general titles and legends must contain all information needed to be explained by themselves.
